# Isolated Aggregators: Towards Forgetting-Free Continual Visual Place Recognition with Fast Adaptation

## Abstract

Visual Place Recognition (VPR), the task of identifying revisited places by a query image, suffers significant degradation in long-term deployment due to non-stationary distribution shifts. Existing methods mainly rely on regularization- and/or replay-based continual learning strategies to address this challenge. However, regularization remains vulnerable to catastrophic forgetting under strong domain shifts, while replay introduces additional storage and latency costs and raises privacy concerns, making online adaptation impractical. To this end, we propose Isolated Aggregators, a new paradigm where each new environment is assigned an independent aggregator following a shared, frozen backbone. By design, parameters for the backbone and all previously learned aggregators are frozen, providing a structural guarantee against catastrophic forgetting. Meanwhile, fine-tuning only the new, lightweight aggregator for the current domain enables fast, privacy-preserving online adaptation to new environments without replay. We further maintain domain descriptors that allow the model to automatically select the appropriate aggregator during inference, ensuring robust continual VPR across diverse environments. Extensive experiments show that our method achieves both zero forgetting and fast adaptation, improving Recall@1 by +9.9% (city-like to nature) and +3.5% (nature to indoor) and within just around 30 and 90 seconds of single-epoch and single-pass training on an NVIDIA RTX 4090. Code will be publicly available.

## 1 Introduction

Visual Place Recognition (VPR) is the task of identifying previously visited locations from a database given a query image. It is a fundamental problem in both computer vision and robotics, supporting core applications such as Structure-from-Motion (SfM) Schönberger & Frahm (2016); Schönberger et al. (2016), visual localization Sarlin et al. (2019), and Simultaneous Localization and Mapping (SLAM) Mur-Artal et al. (2015); Mur-Artal & Tardós (2017). Despite substantial progress in recent years, a critical challenge emerges in real-world deployments, where performance often degrades during domain shifts Lowry et al. (2015), for example when moving from urban to indoor environments, from daytime to nighttime, or across seasons from spring to winter.

Existing work incorporates continual learning into VPR via regularization or replay to address this challenge. An ideal continual VPR system should retain knowledge acquired from previous domains while rapidly adapting to new environments. Regularization-based methods Gao et al. (2022); Ming et al. (2024) penalize updates to parameters deemed important for previously learned domains. However, under strong domain shifts, regularization faces a stability–plasticity dilemma: strong penalties hinder adaptation, whereas relaxed penalties lead to catastrophic forgetting De Lange et al. (2021). Replay-based methods Yin et al. (2023) store and reuse samples from prior domains during training. While effective at mitigating forgetting, replay requires memory that grows with the number of environments, increases data-loading and training latency, which is impractical for resource-constrained robots. It also raises privacy concerns.

Moreover, both methods are less effective with powerful Vision Transformer (ViT) backbones such as DINOv2 Oquab et al. (2023). With transformers, self-attention mixes information across the whole image, making "important" parameters harder to pinpoint. Even tiny updates may spread

widely through the network and cause severe catastrophic forgetting Chefer et al. (2021); Voita et al. (2019). At the same time, stronger backbones improve feature robustness, shifting the main bottleneck from feature extraction to the aggregation module. We find that the aggregation parameters (e.g., soft-assignment score-projection and cluster centroids in NetVLAD Arandjelovic et al. (2016) and score-projection in SALAD Izquierdo & Civera (2024)) remain highly sensitive to domain distribution. Consequently, the key challenge is not only to extract consistent cross-domain features, but also to learn domain-specific aggregations that maintain past-domain performance while adapting quickly to new environments.

Building on these observations, we propose Isolated Aggregators (IA), a framework that freezes a shared backbone and learns an independent, domain-specific aggregator for each new environment. Freezing the backbone preserves base task knowledge and provides a stable feature space for both past and new domains. Adaptation is confined to a new, lightweight aggregator, which is trained from scratch. Since the aggregator contains significantly fewer parameters than the backbone, it converges rapidly, enabling fast and effective adaptation to new environments. To avoid forgetting, we retain all domain-specific aggregators and automatically select the appropriate one at inference. This is achieved by comparing a routing descriptor computed from the current backbone features with each aggregator's learned domain descriptor. The closest match is chosen. By design, our method structurally resolves the stability-plasticity dilemma while incurring substantially lower computational and memory overhead than regularization- or replay-based alternatives, as it updates only a small parameter set and requires no replay buffer. It also avoids the storage growth, latency overhead, and privacy concerns.

In conclusion, our contributions are threefold:

- We identify that for recent VPR models employing powerful foundation backbones like DI-NOv2 Oquab et al. (2023), the continual VPR bottleneck lies in the aggregator rather than the backbone, and we further explain this aggregator sensitivity theoretically in Section 4.1.

- Accordingly, we introduce the first parameter-isolation continual learning framework for VPR, Isolated Aggregators (IA). IA freezes a shared backbone and learns an independent, domain-specific aggregator for each new environment. All aggregators are retained and automatically selected at inference. This design structurally prevents catastrophic forgetting while enabling fast adaptation to new environments.

- We demonstrate through extensive experiments that IA achieves both zero forgetting and rapid adaptation, improving Recall@1 by +9.9% (from city-like to nature) and +3.5% (from nature to indoor) within around 30 s and 90 s of training on an NVIDIA RTX 4090.

## 2 RELATED WORKS

### 2.1 VISUAL PLACE RECOGNITION

The evolution of Visual Place Recognition (VPR) has centered on the aggregator's design, with the consistent goal of converting local/patch features into compact, distinctive global descriptors that remain robust across changing domains Lowry et al. (2015). Classical VPR relied on statistical aggregators to pool hand-crafted local features into a global representation. Prominent example is the Vector of Locally Aggregated Descriptors (VLAD) Jégou et al. (2010), which accumulates the residuals between local features and their assigned visual words.

CNN-based NetVLAD Arandjelovic et al. (2016) is a milestone, which ushered in the data-driven era by introducing a learnable, end-to-end trainable VLAD layer featuring learnable cluster centroids and a soft-assignment mechanism. However, both the learned cluster centroids and the soft-assignment are inherently sensitive to their training distribution, causing significant performance degradation under domain shifts Arandjelovic & Zisserman (2013). Subsequent methods, such as DINOv2-based SALAD Izquierdo & Civera (2024), SuperVLAD Lu et al. (2024) and BoQ Ali-Bey et al. (2024), have attempted to enhance robustness by using various strategies to mitigate the reliance on fixed, learned cluster centroids. Nevertheless, they continue to depend on the domain-sensitive soft-assignment mechanism. This leaves a critical research gap for a long-term, deployable aggregation strategy that is robust by design.

## 2.2 Continual Visual Place Recognition

Continual VPR methods have primarily followed two strategies: regularization and replay. Regularization-based approaches, such as AirLoop Gao et al. (2022) and VIPeR Ming et al. (2024), penalize updates to critical network parameters. While effective for CNNs, these methods fail to adequately constrain modern, large-scale Vision Transformers (ViTs), leading to a poor trade-off between forgetting and adaptation. Replay-based methods, such as BioSLAM Yin et al. (2023), achieve strong memory retention by storing and reusing past data. However, this comes at the cost of high computational overhead, storage requirements, and potential privacy issues, making them impractical for resource-constrained robotic applications. Neither approach offers a scalable continual VPR solution for the era of foundation models, motivating the need for a new paradigm.

## 2.3 Parameter-isolation-based Continual Learning

A third paradigm for continual learning, parameter-isolation, offers an architectural solution to catastrophic forgetting by allocating distinct parameters for each new task. A prominent example of this is Parameter-Efficient Fine-Tuning (PEFT) Houlsby et al. (2019), using techniques like adapters Gao et al. (2024); Zhou et al. (2024) or prompts Wang et al. (2022a;b). These methods freeze the vast majority of the backbone's parameters and insert small, trainable modules, such as adapters between its layers or prompts in the input space, for each new task. While highly successful in general Computer Vision (CV) and Natural Language Processing (NLP) tasks, this paradigm has been underexplored in VPR. Our work, Isolated Aggregators (IA), is the first to apply this parameter-isolation strategy to the VPR aggregator, addressing a key gap in the literature.

## 3 Problem Formulation

**Visual place recognition.** Given a query image $I_q$ and a reference database $\mathcal{D} = \{I_r\}_{r=1}^{N}$, a VPR model maps each image $I$ to a global descriptor $z = f(I) \in \mathbb{R}^d$, which is $\ell_2$-normalized. During training, $f(\cdot)$ is optimized with the multi-similarity loss Wang et al. (2019) over mini-batches labeled by place IDs, pulling same-place pairs together and pushing different-place pairs apart:

$$\mathcal{L}_{MS} = \frac{1}{N} \sum_{i=1}^{N} \left\{ \frac{1}{\alpha} \log \left[ 1 + \sum_{j \in \mathcal{P}_i} e^{-\alpha (s_{ij}-m)} \right] + \frac{1}{\beta} \log \left[ 1 + \sum_{k \in \mathcal{N}_i} e^{\beta (s_{ik}-m)} \right] \right\}, \quad (1)$$

where $\mathcal{P}_i$ and $\mathcal{N}_i$ denote the positive and negative index sets for anchor $i$, respectively; $s_{ij}$ is the cosine similarity between descriptors $z_i$ and $z_j$ (equal to $z_i^\top z_j$ after $\ell_2$ normalization); and $\alpha, \beta$, and $m$ are hyperparameters controlling the sharpness and margin.

**Continual learning setting.** We train across a sequence of environments (domains) $t = 1, \ldots, T$ with non-stationary data. At environment $t$, the available training set is $\mathcal{D}^t = \{(x_j^t, y_j^t)\}_{j=1}^{N_t}$, where each image $x_j^t$ is associated with a place identity $y_j^t \in \mathcal{C}^t$. Images arrive sequentially and each sample is only seen once (single-pass). Data in environment $t$ may induce appearance and condition shifts (e.g., illumination, season, weather). During training at environment $t$, the model has access only to $\mathcal{D}^t$ and labels $y_j^t$; past data and labels $\mathcal{D}^{0:t-1}$ and $\mathcal{C}^{0:t-1}$ are inaccessible.

## 4 Revisiting Modern VPR Pipelines

To map each image to a global descriptor, a single-stage Visual Place Recognition (VPR) model is composed of two main components: a backbone and an aggregator. Commonly used backbones include classical models like ResNet He et al. (2016), as well as the latest Vision Transformers (ViT) Dosovitskiy et al. such as DINOv2 Oquab et al. (2023). The backbone acts as a feature extractor, transforming the raw image into a high-dimensional, spatially rich, and semantically meaningful feature map. The following aggregator processes these feature maps (local descriptors) to produce a compact, highly discriminative global descriptor.

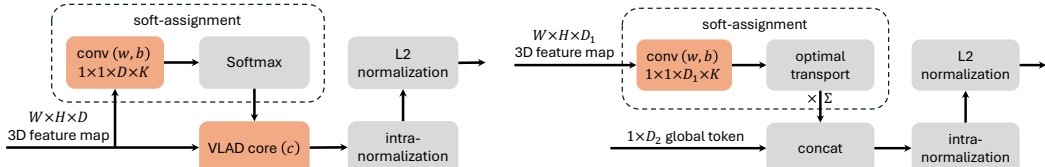

(a) NetVLAD Arandjelovic et al. (2016) aggregator.    (b) SALAD Izquierdo & Civera (2024) aggregator.

Figure 1: Overview of state-of-the-art aggregators in modern VPR pipelines. Orange blocks indicate components with learnable parameters.

## 4.1 AGGREGATORS

**VLAD**    Jégou et al. (2010) aggregates the residuals of each local descriptor with respect to a set of clustered centers. This approach captures richer statistical information about the feature distribution, providing a more robust and discriminative descriptor.

Given $N$ local descriptors $\{x_i\} \in \mathbb{R}^D$, and $K$ cluster centers $\{c_k\}$, the handcrafted VLAD representation is constructed as a $K \times D$ matrix. For each descriptor, the nearest cluster center is identified, and the distance between the descriptor and that center is accumulated into the corresponding column of the matrix. Mathematically, this can be expressed as:

$$V(j,k) = \sum_{i=1}^{N} a_k(x_i)(x_i(j) - c_k(j)), \tag{2}$$

where $a_k(x_i) = 1$ if the descriptor $x_i$ is assigned to cluster $c_k$, and 0 otherwise. Each column therefore encodes the sum of residuals of descriptors associated with one cluster center.

**NetVLAD**    Arandjelovic et al. (2016) extends this concept by introducing a trainable and differentiable version of VLAD. Rather than hard assignment (each local descriptor assigned to its nearest cluster), NetVLAD uses soft assignment, so each local descriptor contributes to multiple clusters with learnable weights produced by a $1 \times 1$ convolution:

$$\bar{a}_k(x_i) = \sum_{i=1}^{N} \frac{e^{w_k^T x_i + b_k}}{\sum_{k'} e^{w_{k'}^T x_i + b_{k'}}}. \tag{3}$$

As shown in Figure 1a, the soft-assignment score-projection weights $w_k$, bias $b_k$, and cluster centers $c_k$ are trainable parameters, allowing the aggregation to be optimized jointly with the backbone through end-to-end learning.

While NetVLAD significantly advanced handcrafted VLAD by enabling supervised, end-to-end training, it remains inherently susceptible to domain shifts. This limitation arises because the parameters learned from the training domain A may not align well with the feature space encountered in a new inference domain B Arandjelovic & Zisserman (2013). This mismatch can be formally described by the residual aggregation in the following equation:

$$V(j,k) = \sum_{i=1}^{N} \frac{e^{w_k^{T,A} x_i^B + b_k^A}}{\sum_{k'} e^{w_{k'}^{T,A} x_i^B + b_{k'}^A}} \left(x_i^B(j) - c_k^A(j)\right), \tag{4}$$

where $w_k^A$, $b_k^A$, and $c_k^A$ are learned from the ***training domain A***, while $x_i^B$ represents the feature maps generated from the backbone in the ***unseen inference domain B***. In this case, the residual aggregation becomes biased toward mismatched centroids, causing the resulting representation to emphasize irrelevant variations. This fundamentally weakens the model's robustness under domain shifts, highlighting a key challenge for real-world, long-term deployment.

**SALAD**    Izquierdo & Civera (2024) mitigates these issues by aggregating features via weighted summation rather than residuals to cluster centroids (see Figure 1b). However, it is not entirely cluster-independent: the score-projection layer used for feature-to-cluster assignment remains sensitive to domain shifts Lu et al. (2024).

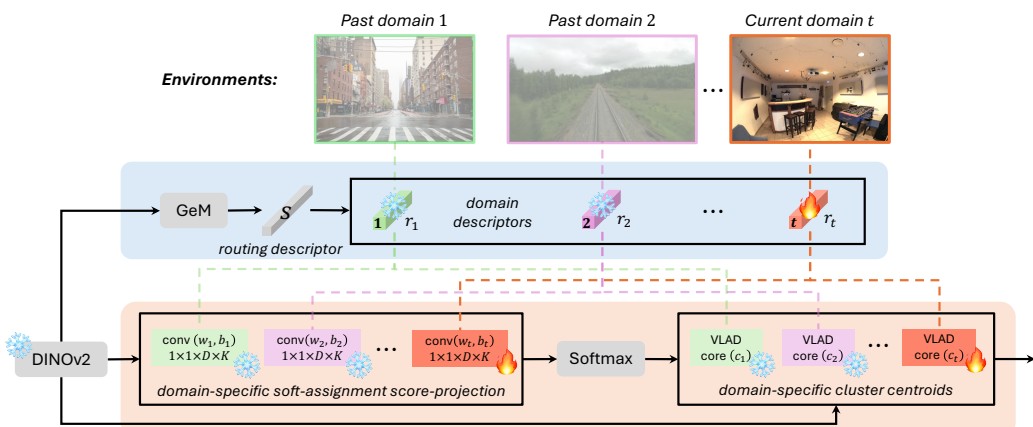

Figure 2: **Overview of our Isolated Aggregators framework with NetVLAD (NetVLAD-IA)**. A frozen DINOv2 backbone produces local descriptors. For each environment, we maintain a lightweight, domain-specific aggregator consisting of a $1 \times 1$ score-projection convolutional layer and cluster centroids. When a new domain $t$ arrives, only the new aggregator is trained; the backbone and all previous aggregators remain frozen. A bank of domain descriptors $\{r_i\}$ is trained and used to choose the appropriate aggregator at inference.

## 5 ISOLATED AGGREGATORS

### 5.1 FAST AND EFFECTIVE DOMAIN ADAPTATION

In Section 4.1, we analyze that NetVLAD and SALAD are vulnerable to domain shifts because their soft-assignment score-projection layer and cluster centroids are learned on the training distribution, leading to misalignment with backbone features extracted in novel environments.

To address this limitation, as shown in Figure 2, we freeze the backbone parameters and learn domain-specific aggregator parameters, i.e., score-projection weights $w_k$, biases $b_k$, and cluster centroids $c_k$, for each newly encountered domain. This ensures that $w_k$, $b_k$, $c_k$, and local descriptors $x_i$ are aligned within the same domain $B$. The VLAD residual for cluster $k$ (at dimension $j$) is then

$$V(j,k) = \sum_{i=1}^{N} \frac{e^{w_k^{T,B} x_i^B + b_k^B}}{\sum_{k'} e^{w_{k'}^{T,B} x_i^B + b_{k'}^B}} \left( x_i^B(j) - c_k^B(j) \right). \tag{5}$$

As a result, the adapted soft-assignment layer can accurately associate local descriptors with the correct clusters, enabling a precise computation of residuals against the correspondingly adapted cluster centroids within the same domain.

### 5.2 TOWARDS FORGETTING-FREE AGGREGATOR ROUTING

A practical challenge at inference is selecting the correct domain-specific aggregator when the current environment is unknown. Assuming access to oracle domain labels is unrealistic, as real-world agents must operate under unseen conditions without annotation. Inspired by ProtoDepth Rim et al. (2025), we introduce a lightweight routing mechanism based on compact *domain descriptors* that automatically selects the appropriate aggregator, enabling forgetting-free aggregator routing without external labels.

**Learning domain descriptors.** During training, each domain $t$ is assigned a learnable descriptor $r_t \in \mathbb{R}^d$. For an input image $x$ from domain $t$, we extract a per-image *routing descriptor* $s(x)$ by global average pooling the backbone bottleneck features (optionally followed by a linear projection) and $\ell_2$-normalize it. Unlike aggregator parameters, $\{r_t\}$ serve only as identifiers for routing; once a domain is learned, its descriptor is frozen. When a new domain arrives, a fresh $r_t$ is initialized

and optimized to align with the current domain while remaining distinct from all previously learned domains. We use a cosine-based objective:

$$\mathcal{L}_D = \big(1 - \cos(\bar{s}_t, r_t)\big) \; + \; \frac{\lambda}{T-1} \sum_{j \neq t} \cos(r_t, r_j), \tag{6}$$

where $\bar{s}_t$ is the mini-batch mean of $s(x)$ for domain $t$, and $\lambda > 0$ balances alignment and separation. The overall training loss combines VPR training with descriptor learning:

$$\mathcal{L} \; = \; \mathcal{L}_{MS} \; + \; \mathcal{L}_D. \tag{7}$$

**Routing at inference.** Given a query image, we compute its routing descriptor $s$ and select the aggregator whose descriptor is most similar:

$$t^* \; = \; \arg\max_t \; \cos(s, r_t). \tag{8}$$

This routing is efficient, deterministic, and label-free. It scales gracefully as new environments are introduced, avoids catastrophic forgetting by never modifying past aggregators, and activates only the relevant expert at inference, providing a practical and robust solution for lifelong VPR under continuously evolving conditions. The effectiveness of aggregator routing via domain descriptors is evaluated in Section A.3.

# 6 EXPERIMENTS

## 6.1 IMPLEMENTATION DETAILS

We evaluate our Isolated Aggregators (IA) with two state-of-the-art aggregators, NetVLAD Arandjelovic et al. (2016) and SALAD Izquierdo & Civera (2024), on a DINOv2 Oquab et al. (2023) backbone. To ensure fairness, we adopt the same aggregator configurations and optimization settings as NetVLAD and SALAD. Experiments are conducted on an NVIDIA RTX 4090 GPU.

## 6.2 BASELINES

We compare IA against the regularization-based AirLoop Gao et al. (2022) and replay via Experience Replay (ER) Rolnick et al. (2019). For reference, we also report a non-continual baseline: full fine-tuning on each environment (*Finetuned*). We attempted to include BioSLAM Yin et al. (2023) and VIPeR Ming et al. (2024), but their implementations are closed-source at the time of writing, preventing reproduction.

## 6.3 CONTINUAL PROTOCOLS AND EVALUATION DATASETS

We categorize VPR domain shifts into two types and design corresponding continual learning protocols: (i) Location-induced shifts, which occur when transitioning between geographical contexts (e.g., urban → natural → indoor); and (ii) Condition-induced shifts, which arise from appearance variations like changes in illumination, weather, or season.

**Location-induced protocol.** To the best of our knowledge, we are the first to propose this challenging cross-location protocol for continual VPR evaluation. In this protocol, we start from a base model pre-trained on GSV-Cities Ali-bey et al. (2022) (city-like) and then adapt and evaluate it sequentially on Nordland Olid et al. (2018) (natural) and ScanNet Dai et al. (2017) (indoor). The evaluation datasets are described in Section A.1.

**Condition-induced protocol.** Following Gao et al. (2022), we also evaluate condition-induced shifts on Nordland. Unlike their setups, we initialize with a base model pre-trained on the GSV-Cities dataset before adapting to new conditions. The model is adapted sequentially across the four seasons (spring, summer, fall, winter) and evaluated after each step.

To mimic realistic deployment, all adaptation is performed in a single epoch with a single pass over each image. Past data are not accessible (no replay).

## 6.4 EVALUATION METRICS

To evaluate the model's ability to retain past knowledge and adapt to new domains, we adopt the evaluation protocol in previous work Gao et al. (2022). We form a $T \times T$ matrix $R$, where $R_{i,j}$ is the R@1 on environment $j$ after training on environment $i$. We report three scalars: Average Performance (AP), which provides a holistic view of the model's effectiveness; Backward Transfer (BWT), where a negative value signifies catastrophic forgetting. The metrics are calculated as:

$$\text{AP} = \frac{\sum_{i=1}^{T} \sum_{j=1}^{i} R_{i,j}}{T(T+1)/2}, \quad \text{BWT} = \frac{\sum_{i=2}^{T} \sum_{j=1}^{i-1} (R_{i,j} - R_{j,j})}{T(T-1)/2}. \tag{9}$$

Table 1: Our Isolated Aggregation (IA) framework achieves strong adaptation and zero-forgetting capacity in both location- and condition-induced continual protocols. ↑ denotes improvement over the base or seen domain on the same dataset; → denotes no forgetting relative to the previous step.

(a) Location-induced continual VPR. Starting from a base model pre-trained on a city-like dataset Ali-bey et al. (2022), the model is adapted sequentially to *Nordland* Olid et al. (2018) and then *ScanNet* Dai et al. (2017). *Base* reports performance before adaptation. Under *IA (Ours)*, the *Nordland* column is performance immediately after adapting to Nordland; the *ScanNet* column is performance after subsequently adapting to ScanNet.

| Aggregator | Dataset | Base | IA (Ours) | |
| --- | --- | --- | --- | --- |
| | | | Nordland | ScanNet |
| NetVLAD | Nordland | 70.7 | 74.5 ↑ | 74.5 → |
| | ScanNet | 90.5 | 90.5 | 90.5 → |
| SALAD | Nordland | 76.6 | 86.5 ↑ | 86.5 → |
| | ScanNet | 88.5 | 88.0 | 91.5 ↑ |

(b) Condition-induced continual VPR on Nordland Olid et al. (2018). Starting from a city-like pre-trained base, the model is adapted sequentially across seasons (spring → summer → fall → winter). Each row indicates the *evaluation season*; *Base* is performance before any seasonal adaptation. Under *IA (Ours)*, columns report performance after each adaptation step.

| Aggregator | Season | Base | IA (Ours) | | | |
| --- | --- | --- | --- | --- | --- | --- |
| | | | spring | summer | fall | winter |
| NetVLAD | spring | 73.7 | 74.4 ↑ | 74.4 → | 74.4 → | 74.4 → |
| | summer | 70.8 | 71.4 | 71.5 ↑ | 71.5 → | 71.5 → |
| | fall | 70.6 | 71.3 | 71.5 | 71.4 ↑ | 71.4 → |
| | winter | 72.0 | 72.5 | 72.5 | 72.5 | 72.6 ↑ |
| SALAD | spring | 72.7 | 76.1 ↑ | 76.7 ↑ | 76.6 ↑ | 76.6 ↑ |
| | summer | 70.9 | 73.9 | 74.9 ↑ | 75.1 ↑ | 75.1 ↑ |
| | fall | 70.7 | 73.4 | 74.1 | 74.4 ↑ | 74.4 → |
| | winter | 69.2 | 71.7 | 71.7 | 71.7 | 72.3 ↑ |

## 6.5 CONTINUAL PLACE RECOGNITION PERFORMANCE

### 6.5.1 LOCATION-INDUCED SHIFTS

We first evaluate our Isolated Aggregators (IA) framework on the challenging location-induced continual learning protocol, with results presented in Table 1a. Our method demonstrates both strong forward adaptation and perfect backward retention (zero forgetting).

NetVLAD-IA improves Recall@1 by 3.8% (from 70.7% to 74.5%) when adapting from the urban base domain to the natural Nordland dataset. As reported in the table, when the model subsequently adapts to the indoor domain (ScanNet), it perfectly retains its performance on Nordland (74.5%), achieving zero forgetting.

The improvements are even more pronounced with the stronger SALAD aggregator. SALAD-IA achieves a remarkable +9.9% (76.6% to 86.5%) when adapting to Nordland. Furthermore, when transitioning to the indoor ScanNet dataset, it not only maintains its expert performance on Nordland (86.5%) but also continues to improve its performance by 3.5%, demonstrating effective adaptation.

As shown in Table 2, IA outperforms baseline continual VPR methods on both overall performance (AP) and forgetting (BWT). With the NetVLAD aggregator, IA attains the best AP of 79.8%, exceeding AirLoop and Replay by 12.6% and 7.5%, respectively. It is also the only method with a positive BWT (+1.3%), indicating no catastrophic forgetting, whereas AirLoop and Replay exhibit substantial forgetting (BWT of -6.5% and -2.7%, respectively). The advantage is even more significant with the stronger SALAD aggregator: SALAD-IA achieves an AP of 88.2% and a BWT of +4.3%, outperforming all baselines.

Table 2: Continual VPR performance comparison in both location-induced and condition-induced protocols. IA demonstrates a significant advantage in both overall performance (AP) and catastrophic forgetting (BWT) abilities over existing continual learning strategies.

| Aggregator | Method | Location-induced | | Condition-induced | |
|---|---|---|---|---|---|
| | | AP | BWT | AP | BWT |
| NetVLAD | Finetuned | 59.2 | -18.1 | 51.6 | -0.2 |
| | AirLoop Gao et al. (2022) | 67.2 | -6.5 | 62.9 | -0.1 |
| | Replay Rolnick et al. (2019) | 72.3 | -2.7 | 66.2 | **0** |
| | **IA (Ours)** | **79.8** | **+1.3** | **72.8** | **0** |
| SALAD | Finetuned | 62.1 | -18.5 | 53.0 | -0.1 |
| | AirLoop Gao et al. (2022) | 60.6 | -10.1 | 51.4 | **+0.7** |
| | Replay Rolnick et al. (2019) | 76.1 | -2.3 | 67.3 | +0.1 |
| | **IA (Ours)** | **88.2** | **+4.3** | **75.2** | +0.3 |

### 6.5.2 CONDITION-INDUCED SHIFTS

In the condition-induced protocol on the Nordland dataset, our IA framework demonstrates a strong ability to continuously adapt to seasonal changes while perfectly preserving knowledge from previously seen conditions. The results are detailed in Table 1b.

With NetVLAD as the base aggregator, IA shows consistent positive adaptation. For example, when first adapting to 'spring', performance improves from 73.7% to 74.4%. As the model sequentially adapts to 'summer', 'fall', and 'winter', it not only learns the new conditions but also maintains its peak performance on all previously seen seasons, achieving zero forgetting in all subsequent steps.

Using the SALAD aggregator, the benefits of our framework are even more evident. SALAD-IA shows significant and cumulative performance gains across the seasonal sequence. It achieves a 3.4% gain on 'spring' (72.7% to 76.1%), and continues to improve its performance on both 'spring' and 'summer' even as it adapts to 'fall'. This demonstrates strong adaptation on unseen environments, while achieving zero forgetting on past domains.

Additionally, our IA framework continues to lead in overall performance while completely eliminating forgetting, as reported in Table 2. For both NetVLAD and SALAD, IA achieves the highest AP (72.8% and 75.2%, respectively). In terms of forgetting, our method achieves a BWT of 0 with NetVLAD and +0.3% with SALAD, effectively achieving zero-forgetting capability. Notably, IA achieves this perfect retention while delivering a significantly higher AP than Replay and AirLoop, confirming its ability to adapt to new environments without forgetting past knowledge.

We report an additional condition-induced shift experiment on the RobotCar Maddern et al. (2017) dataset in Section A.2, where illumination changes follow sun → night → overcast. The results further show that our Isolated Aggregators (IA) framework is markedly more stable than regularization Gao et al. (2022) under condition-induced shifts against catastrophic forgetting.

### 6.6 ONLINE LEARNING RUNTIME PERFORMANCE

We report the online learning runtime performance of our method on both the Nordland and ScanNet datasets. All runtime experiments are conducted on an NVIDIA RTX 4090 GPU, and each value is the average of 10 runs.

As reported in Table 3a, both NetVLAD-IA and SALAD-IA achieve efficient online adaptation. On the Nordland dataset, with 4,096 places, NetVLAD-IA requires 90.33 seconds, while SALAD-IA completes training in 87.54 seconds. On the ScanNet dataset, which contains 1,513 indoor scenes, NetVLAD-IA finishes in 29.84 seconds, and SALAD-IA achieves a slightly faster runtime of 27.1 seconds. In each place and scene, we randomly select 4 images for training.

These results demonstrate that our continual learning framework not only provides strong adaptation and robustness against forgetting, as shown in previous experiments, but also maintains high computational efficiency during online training, making it practical for real-time deployment.

Table 3: Runtime performance and memory footprint of our Isolated Aggregators (IA) framework.

(a) Runtime performance of our IA.

| Dataset | #places | NetVLAD-IA | SALAD-IA |
|---------|---------|------------|----------|
| Nordland | 4,096 | 90.33s | 87.54s |
| ScanNet | 1,513 | 29.84s | 27.1s |

(b) Memory footprint of our NetVLAD-IA framework.

| Object | #parameters | float64 (8B) | float32 (4B) | float16 (2B) | int8 (1B) |
|--------|-------------|--------------|--------------|--------------|-----------|
| Score-projection weights | 24,576 | 192 KB | 96 KB | 48 KB | 24 KB |
| Cluster centroids | 24,576 | 192 KB | 96 KB | 48 KB | 24 KB |

## 6.7 MEMORY FOOTPRINT ANALYSIS

We further report the memory overhead of our method in Table 3b. For NetVLAD-IA, we store the soft-assignment score-projection weights (192 KB) and cluster centroids (192 KB), totaling 384 KB per domain. The cost scales linearly with the number of domains: e.g., 10 domains for 3.75 MB, 1,000 domains 375 MB. For SALAD-IA, only the score-projection layer is stored, i.e., 192 KB per domain (10 domains 1.875 MB; 1,000 domains 187.5 MB).

These lightweight footprints make IA highly scalable and memory-efficient, supporting long-term continual operation across many environments without prohibitive storage.

Table 4: Effect of architectural choice for adaptation on continual performance.

| Aggregator | Dataset | Base | Configuration I | | Configuration II | | IA (Ours) | |
|------------|---------|------|----------|---------|----------|---------|----------|---------|
| | | | Nordland | ScanNet | Nordland | ScanNet | Nordland | ScanNet |
| NetVLAD | Nordland | 70.7 | 73.5 ↑ | 73.2 ↓ | 50.9 ↓ | 43.5 ↓ | **74.5** ↑ | **74.5** → |
| | ScanNet | 90.5 | 90.5 | 90.5 → | 88.5 | **97.0** ↑ | 90.5 | 90.5 → |
| SALAD | Nordland | 76.6 | 83.8 ↑ | 83.2 ↓ | 42.1 ↓ | 29.8 ↓ | **86.5** ↑ | **86.5** → |
| | ScanNet | 88.5 | 88.5 | 90.5 ↑ | 86.5 | **96.0** ↑ | 88.0 | 90.5 ↑ |

## 6.8 ABLATION STUDY

In this section, we examine how the adaptation locus affects continual performance (Table 4). Configuration I (frozen backbone with shared-aggregator adaptation) delivers notable adaptation gains but introduces forgetting on the previously learned domain: with NetVLAD, performance on Nordland drops by 0.3% after adapting to ScanNet; with SALAD, a similar decline is observed (-0.6%). By contrast, our isolated aggregators (IA) framework achieves both stronger adaptation (+3.8% and +9.9% on Nordland for NetVLAD and SALAD, respectively, versus +2.8% and +7.2% under Configuration I) and forgetting prevention, showing no forgetting on the previously learned domain.

Configuration II (backbone adaptation with a frozen shared aggregator) yields the largest gains on ScanNet, but at the cost of catastrophic forgetting on Nordland. For NetVLAD, it drops by 19.8% and 27.2% after sequential adaptation. For SALAD, the collapse is even larger. This further demonstrates IA's effectiveness at preventing forgetting while enabling strong adaptation.

## 7 CONCLUSION

We presente *Isolated Aggregators (IA)*, a continual VPR framework that freezes a shared backbone and learns lightweight, domain-specific aggregators. Across cross-location and cross-condition protocols, IA achieves effective adaptation within tens of seconds while preserving prior-domain performance, yielding zero forgetting. Our study offers two key insights. First, after sufficient base-domain pretraining with a strong backbone (e.g., DINOv2), adapting the *aggregator* yields larger gains than updating backbone parameters under substantial domain shifts. Second, this behavior arises from a distribution mismatch between backbone features and the aggregator's statistics (soft-assignment projections and cluster centroids); IA addresses this by isolating parameters so that descriptors, projections, and centroids are aligned within each domain, thereby eliminating interference with previously learned domains. Although effective, IA currently relies on prior information to detect domain changes during training. Future work includes automatic change detection. Overall, IA reframes where and how adaptation should occur in continual VPR, moving the field toward forgetting-free, fast adaptation and deployment-ready lifelong operation.

**Reproducibility statement.** We describe the base encoder and aggregators (NetVLAD and SALAD) together with all training settings in Section 6.1, our proposed method in Section 5, the evaluation protocols and datasets in Section 6.3, and metrics in Section 6.4. The code and evaluated datasets will be released publicly upon acceptance.

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

## A APPENDIX

### A.1 EVALUATION DATASETS

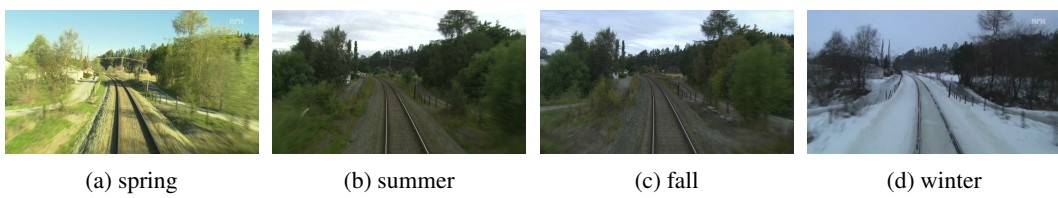

|(a) spring|(b) summer|(c) fall|(d) winter|

Figure 3: Visualization of samples from the Nordland Olid et al. (2018) dataset.

**Nordland.** The Nordland Olid et al. (2018) dataset captures a long-distance train journey across Norway, recorded once per season (summer, fall, winter, spring), resulting in identical routes under dramatically different seasonal appearances. The training split uses two route sections with 24,570 images per season. During training, we group every three consecutive frames into a single place ID and remove adjacent overlaps to avoid place aliasing.

For the location-induced protocol, Nordland (Test) contains 2,760 query images from summer and 27,592 reference images from winter; a match within $\pm 1$ frame is considered ground truth. For the condition-induced protocol, Nordland (Test) uses the full set of 27,592 images from each of the four seasons as queries and references, respectively, with the same $\pm 1$-frame ground-truth criterion. A visualization is shown in Figure 3.

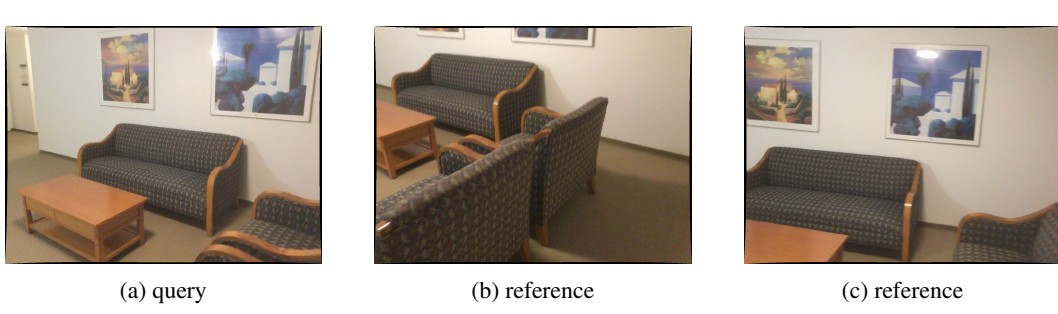

(a) query      (b) reference      (c) reference

Figure 4: Visualization of samples from the ScanNet Dai et al. (2017) dataset.

**ScanNet.** To the best of our knowledge, we are the first to use ScanNet Dai et al. (2017) to evaluate indoor VPR. From the 1,513 indoor scenes, we randomly sample 12 images per scene for training. For evaluation, we select 100 scenes disjoint from the training set and, for each scene, sample 2 query images and 20 reference images to form the database. A visualization is shown in Figure 4.

**RobotCar.** We follow the same setting as AirLoop Gao et al. (2022), which selects three environments based on the lighting condition, labeled sun, overcast, and night. For each environment, we select two sequences as the training and test set, respectively. Ground truth is defined as query–reference pairs with distance $< 10$ m and yaw difference $< 15°$. A visualization is shown in Figure 5.

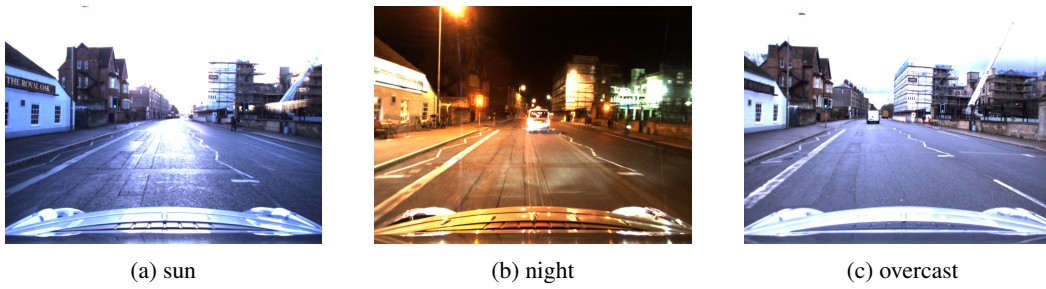

| (a) sun | (b) night | (c) overcast |

Figure 5: Visualization of samples from the RobotCar Maddern et al. (2017) dataset.

## A.2  ADDITIONAL CONTINUAL PLACE RECOGNITION PERFORMANCE

Table 5: Condition-induced continual VPR on RobotCar Maddern et al. (2017). Starting from a city-like pre-trained base, the model is adapted sequentially across illumination (sun → night → overcast). Each row indicates the *evaluation illumination*; *Base* is performance before any illumination adaptation. Under *IA (Ours)*, columns report performance after each adaptation step.

| Seq. | NetVLAD | NetVLAD-AirLoop | | | NetVLAD-IA (Ours) | | |
|---|---|---|---|---|---|---|---|
| | | sun | night | overcast | sun | night | overcast |
| sun | 71.5 | 49.5 ↓ | 52.9 ↑ | 49.7 ↑ | 71.5 → | 71.6 ↑ | 71.6 ↑ |
| night | 72.3 | 48.2 | 51.5 ↓ | 47.4 ↓ | 72.3 | 72.4 ↑ | 72.5 ↑ |
| overcast | 69.4 | 52.8 | 54.6 | 49.7 ↓ | 69.4 | 69.5 | 69.5 ↑ |

Table 5 shows performance when adapting sequentially across illumination (sun → night → overcast). With NetVLAD, IA preserves or slightly improves the base score at every step, e.g., evaluating on sun: 71.5 → 71.5 → 71.6 → 71.6; night: 72.3 → 72.3 → 72.4 → 72.5; on overcast: 69.4 → 69.4 → 69.5 → 69.5, indicating zero forgetting and small positive backward transfer. In contrast, AirLoop, which is strong under standard continual settings, suffers large drops immediately after the first adaptation (e.g., sun: 71.5 → 49.5, night: 72.3 → 51.5, overcast: 69.4 → 49.7) and never recovers to the base, yielding consistently negative BWT. These results highlight that our IA is markedly more stable than regularization for condition-induced shifts against catastrophic forgetting.

## A.3  STUDY ON THE EFFECTIVENESS OF AGGREGATOR ROUTING

Table 6: Domain recognition accuracy via aggregator routing.

| Method | Nordland | | RobotCar | | | Location-induced |
|---|---|---|---|---|---|---|
| | summer | winter | sun | night | overcast | |
| NetVLAD-IA | 88.5 | 94.3 | 98.7 | 98.1 | 95.6 | 94.3 |
| SALAD-IA | 90.3 | 95.1 | 98.3 | 97.6 | 95.8 | 95.6 |

As reported in Table 6, routing accuracy is consistently high across datasets. On Nordland (seasonal shifts), accuracy is 91.4% for NetVLAD-IA (88.5% summer / 94.3% winter) and 92.7% for SALAD-IA (90.3% / 95.1%). The lower accuracy for the summer stems from their visual similarity: as shown in Figures 3a, 3b, and 3c, summer exhibit only subtle appearance differences with spring and fall, which leads to confusion among them. In contrast, winter (Figure 3d) presents a much larger appearance shift, resulting in higher recognition accuracy. On RobotCar (illumination shifts), routing is near-perfect: 97.5% for NetVLAD-IA (98.7% sun, 98.1% night, 95.6% overcast) and 97.2% for SALAD-IA (98.3%, 97.6%, 95.8%). For the location-induced setting, accuracy is 94.3% (NetVLAD-IA) and 95.6% (SALAD-IA). Overall averages are 94.9% and 95.8% for NetVLAD-IA and SALAD-IA, respectively, indicating reliable routing via domain descriptors.

### A.4 The Use of Large Language Models (LLMs)

We used ChatGPT and Gemini solely for language polishing.

