# OpenReview forum: "Isolated Aggregators: Towards Forgetting-Free Continual Visual Place Recognition with Fast Adaptation"
_ICLR.cc/2026/Conference — ICLR 2026 Conference Withdrawn Submission_

### Official Review · Reviewer_Dwgh · 2025-10-28

**Soundness:** 2
**Presentation:** 2
**Contribution:** 1
**Rating:** 2
**Confidence:** 5

**Summary:**

The paper proposes Isolated Aggregators (IA), a parameter-isolation strategy for continual Visual Place Recognition (VPR), where each environment is assigned its own aggregator, while a shared backbone remains frozen. The claimed benefit is fast adaptation and zero forgetting without replay or regularization.

While the idea is simple and somewhat intuitive, the paper falls short in terms of novelty, rigor, and empirical depth. Overall, the contribution feels incremental and primarily an engineering variation on well-known modular or adapter-based continual learning strategies.

**Strengths:**

1. The paper is well-written and easy to follow, with structured sections and clear illustrations.
2. Addressing continual domain shifts in VPR is a relevant and challenging problem.
3. The authors evaluate across multiple datasets (Nordland, ScanNet, RobotCar) with reasonable baselines.

**Weaknesses:**

1. The proposed “Isolated Aggregators” are conceptually equivalent to task-specific heads or adapters, a well-established idea in continual learning (e.g., PEFT, adapters, or expert routing). The method merely applies this known paradigm to VPR, without introducing a fundamentally new mechanism or theory.

2. The “forgetting-free” claim is largely tautological since freezing parameters trivially prevents forgetting; this does not address lifelong learning scalability or domain-overlap issues. The “fast adaptation” results are expected given that only small layers are trained.

3. Baselines (AirLoop, Replay) are insufficient; state-of-the-art continual learning methods (e.g., prompt-based, parameter-isolation models, or memory-efficient PEFTs) are missing.

**Questions:**

None.

---

### Official Review · Reviewer_djRg · 2025-10-31

**Soundness:** 1
**Presentation:** 1
**Contribution:** 2
**Rating:** 4
**Confidence:** 4

**Summary:**

This paper proposes a approach for VPR under continual learning scenarios, where a system must adapt to new environments without forgetting previously learned knowledge. The method, named Isolated Aggregators, addresses the limitations of regularization and replay-based strategies. It freezes the backbone network and learns lightweight, domain-specific aggregators for each environment, which are adapted independently while preserving past knowledge. The aggregators are selected dynamically at inference using domain descriptors, ensuring fast, efficient, and privacy-preserving adaptation to new domains. The proposed method shows zero forgetting and fast adaptation in extensive experiments across various environments, with substantial improvements in recognition performance.

**Strengths:**

1. The paper provides a solution to the catastrophic forgetting problem by isolating domain-specific aggregators, which significantly reduces memory overhead and computational costs.
2. The method operates without needing to store and replay past data, addressing the privacy concerns and computational burdens typical of replay-based methods.

**Weaknesses:**

1. The paper proposes using domain descriptors to guide the selection of appropriate aggregators during inference. However, while the concept is promising, the robustness of these domain descriptors in more challenging or diverse real-world environments is not fully explored. For example, in cases where environmental conditions are drastically different, such as extreme weather or highly dynamic scenes, it is unclear how well the domain descriptors will generalize. The paper could benefit from experiments that demonstrate the performance of the routing system under more extreme or less controlled conditions, particularly for real-world deployments.
2. While the experiments conducted in the paper are thorough, the evaluation metrics could be expanded to include a more diverse set of measures that are critical for deployment in real-world applications. Specifically, the current metrics (Recall@1, Average Performance (AP), and Backward Transfer (BWT)) provide insights into the model’s ability to adapt to new environments while retaining past knowledge. However, real-time performance metrics, such as computational cost, memory usage, or latency during inference, especially when dealing with large-scale environments or when continuously adapting to numerous environments, could offer more practical insights.
3. Although the paper highlights the scalability of the method with respect to memory usage per domain, it does not fully explore the scalability when the number of environments grows significantly. For instance, while the method shows memory efficiency for 10 or 1000 domains, it is not clear how the system will behave as the number of domains continues to grow beyond these examples, especially when considering more complex, diverse environments.
4. The paper mentions that the aggregators are lightweight and trained from scratch for each new domain, but it does not provide a detailed analysis of the trade-offs between aggregator size and model performance. For instance, there may be a point at which the size of the aggregator (number of parameters) significantly impacts the performance or computational efficiency. While the lightweight nature of the aggregators is an advantage, the paper could delve deeper into whether there are diminishing returns as the aggregator size increases.
5. One weakness of the paper is the lack of visualizations that would help clarify the proposed method and experimental results.

**Questions:**

1. The paper proposes using domain descriptors to guide the selection of appropriate aggregators during inference. However, while the concept is promising, the robustness of these domain descriptors in more challenging or diverse real-world environments is not fully explored. For example, in cases where environmental conditions are drastically different, such as extreme weather or highly dynamic scenes, it is unclear how well the domain descriptors will generalize. The paper could benefit from experiments that demonstrate the performance of the routing system under more extreme or less controlled conditions, particularly for real-world deployments.
2. While the experiments conducted in the paper are thorough, the evaluation metrics could be expanded to include a more diverse set of measures that are critical for deployment in real-world applications. Specifically, the current metrics (Recall@1, Average Performance (AP), and Backward Transfer (BWT)) provide insights into the model’s ability to adapt to new environments while retaining past knowledge. However, real-time performance metrics, such as computational cost, memory usage, or latency during inference, especially when dealing with large-scale environments or when continuously adapting to numerous environments, could offer more practical insights.
3. Although the paper highlights the scalability of the method with respect to memory usage per domain, it does not fully explore the scalability when the number of environments grows significantly. For instance, while the method shows memory efficiency for 10 or 1000 domains, it is not clear how the system will behave as the number of domains continues to grow beyond these examples, especially when considering more complex, diverse environments.
4. The paper mentions that the aggregators are lightweight and trained from scratch for each new domain, but it does not provide a detailed analysis of the trade-offs between aggregator size and model performance. For instance, there may be a point at which the size of the aggregator (number of parameters) significantly impacts the performance or computational efficiency. While the lightweight nature of the aggregators is an advantage, the paper could delve deeper into whether there are diminishing returns as the aggregator size increases.
5. One weakness of the paper is the lack of visualizations that would help clarify the proposed method and experimental results.

---

### Official Review · Reviewer_ikgZ · 2025-11-01

**Soundness:** 2
**Presentation:** 2
**Contribution:** 2
**Rating:** 4
**Confidence:** 5

**Summary:**

This paper proposes Isolated Aggregators (IA), a continual learning framework for Visual Place Recognition (VPR) that freezes a shared backbone and trains lightweight, domain-specific aggregators for each new environment. The method aims to achieve zero catastrophic forgetting and fast adaptation by isolating aggregator parameters and using a routing mechanism based on domain descriptors. The authors demonstrate strong performance on location- and condition-induced domain shifts using datasets such as Nordland, ScanNet, and RobotCar. The approach is shown to be computationally efficient, requiring only tens of seconds for adaptation on an NVIDIA RTX 4090.

**Strengths:**

The idea of isolating aggregators to prevent catastrophic forgetting is novel and well-motivated, especially in the context of modern VPR pipelines with powerful backbones like DINOv2.

The method achieves impressive adaptation speed and zero forgetting in the evaluated scenarios, with clear experimental validation and ablation studies.

**Weaknesses:**

1. The experimental validation lacks comparisons with several important VPR methods, such as BoQ[1], CricaVPR[2], and SelaVPR[3], making it difficult to assess the true advancement of the proposed method over the current state-of-the-art.

2. The evaluation is conducted on a limited set of datasets and crucially omits common VPR benchmarks like Pitts30k, MSLS, and Tokyo24/7, which severely undermines the claims of generalizability and makes comparison with the broader VPR literature impossible.

3. The paper does not discuss the system-level scalability when many domains are accumulated, leaving the computational overhead of the routing mechanism and the potential for descriptor interference in large-scale lifelong learning scenarios unaddressed.

4. There is no in-depth analysis of the failure modes of the critical routing mechanism, such as how often routing errors occur or the direct impact of an incorrect aggregator selection on the final VPR performance.

5. The method assumes discrete domain shifts during training and does not address how it would handle gradual, smooth domain transitions where clear boundaries for initializing a new aggregator do not exist, which is a common real-world challenge.

References

[1] Boq: A place is worth a bag of learnable queries. CVPR 2024.

[2] Cricavpr: Cross-image correlation-aware representation learning for visual place recognition. CVPR 2024.

[3] Towards seamless adaptation of pre-trained models for visual place recognition. ICLR 2024.

**Questions:**

Please see weaknesses.

---

### Note · Authors · 2025-11-19

I have read and agree with the venue's withdrawal policy on behalf of myself and my co-authors.